# Lithium Medication in Pregnancy and Breastfeeding—A Case Series

**DOI:** 10.3390/medicina57060634

**Published:** 2021-06-18

**Authors:** Andrea Gehrmann, Katrin Fiedler, Anna Linda Leutritz, Carolin Koreny, Sarah Kittel-Schneider

**Affiliations:** 1Department of Psychiatry, Psychotherapy and Psychosomatic Medicine, University Hospital, University of Würzburg, D-97080 Würzburg, Germany; Gehrmann_A1@ukw.de (A.G.); Leutritz_A@ukw.de (A.L.L.); carolin.koreny@gmail.com (C.K.); 2Department of Psychiatry, Psychotherapy and Psychosomatic Medicine, University Hospital, Goethe-University, D-60528 Frankfurt, Germany; Katrin.fiedler@kgu.de

**Keywords:** lithium, pregnancy, lactation

## Abstract

Lithium salts are the first-line prophylaxis treatment for bipolar disorder in most guidelines. The majority of bipolar women are treated with mood stabilizers at the time they wish to get pregnant. One reason for this is the rising average age at first childbirth, at least in the high-income countries, which increases in general the likelihood of a medication with psychotropic drugs. Previously, lithium exposition during pregnancy was thought to strongly increase the risk of severe cardiac malformation. However, recent studies only point to a low teratogenic risk, so nowadays an increasing number of women are getting pregnant with ongoing lithium treatment. Regarding lithium medication during breastfeeding, there is evidence that lithium transfers to the breastmilk and can also be detected in the infants’ serum. The influence on the infant is still a largely understudied topic. Regular monitoring of the infants’ renal clearance, thyroid function, and lithium levels is warranted when breastfeeding under lithium exposure. In this case series, we present three case reports of bipolar mothers who were treated with lithium during pregnancy and breastfeeding to add to the scarce literature on this important topic. In short, we strengthen the importance of therapeutic drug monitoring due to fluctuating plasma levels during pregnancy and after birth, and we can report the birth and development of three healthy infants despite lithium medication during pregnancy and breastfeeding.

## 1. Introduction

Bipolar disorder affects 1–2% of the adult population worldwide and has a similar prevalence in women and men [1,2]. The typical age of onset is in adolescence and early adulthood [1]. Because of the increasing mean age of primiparous women in the developing countries, there is a rising number of bipolar women who are already diagnosed and treated with mood stabilizers and wish to become pregnant (for an overview of the increasing mean age at birth in high to middle income countries from 1960 to 2019 see https://www.humanfertility.org/cgi-bin/main.php, assessed on 1 May 2021). While for example quetiapine, which is also widely used as a mood stabilizer in bipolar disorder, is considered among the safest options in pregnancy and lactation [3], lithium as a general first-line treatment in most guidelines on bipolar disorders, has several risks associated with pregnancy and lactation use. However, recent studies could show a much lower risk of cardiac or general malformation than previously thought [4,5]. Additionally, the teratogenic risk seems to be positively correlated with lithium concentration during pregnancy [6]. Furthermore, there is data from a small study showing no negative long-term effect in lithium-exposed children of up to 15 years [7]. Even lesser data is available on lithium medication during breastfeeding. However, breastmilk is the best nutrition for infants in the first year and has several beneficial effects on the health of children as well as the mothers [8,9,10,11,12]. This has to be weighed against the potential harmful effects of lithium exposure. In addition, it has been consistently shown that the postpartum period is an extremely vulnerable period for relapse in bipolar women. Continuing the mood stabilizing medication with lithium could reduce the risk of postpartum relapse substantially. In one retrospective study, only 24% of bipolar women who continued lithium relapsed compared with 70% who discontinued lithium [13]. A severe mood episode in early infancy can negatively impact mother-to-child bonding, mother–child interaction and the development of the child, accordingly it is very important to reduce the relapse risk as best as possible [14,15]. However, lithium is excreted in the breastmilk and can be found in varying degrees between 0 and 50% of the mother’s serum levels [16]. While a recent review of the published data shows that the majority of breastfed infants do not display negative effects, there are still concerns also due to the scarcity of data [16]. Poels and colleagues, therefore, discourage breastfeeding under lithium medication as a conclusion of their systematic review [16], while Pacchiarotti et al. concluded that breastfeeding could be possible because in the available data no severe negative effects have been reported in the exposed infants [17]. Additionally, Hermann et al. suggested that lithium and breastfeeding should not be mutually exclusive, but close collaboration with a pediatrician and regular monitoring and measuring of the exposed infant’s blood lithium level, renal function, and thyroid function are recommended [18]. Here, we report three additional cases of lithium therapy in pregnancy and breastfeeding to add to the published evidence.

## 2. Cases

We present three cases of women with bipolar affective disorder who were treated in our specialized outpatient clinics at the University Hospital of Würzburg and Frankfurt. Those women were treated with long-acting lithium carbonate during pregnancy and breastfeeding and we collected the clinical data from mothers as well as babies during the first months after birth. The patients gave written consent for publishing their cases and/or were participating in the therapeutic drug monitoring breastmilk and gave informed written consent for this. The study protocol was approved by the ethics committee of the University Hospital of Frankfurt (votum no. 136/17).

Lithium levels and renal function values were measured from serum from routine blood parameter checks in the central laboratory of the University Hospital Würzburg and Frankfurt. Case 1 was taking Hypnorex ret.^®^, which is manufactured by Essential Pharma Ltd. The other two patients were taking Quilonum^®^ ret., which is either manufactured by Teofarma S.R.L. or GlaxoSmith Kline.

### 2.1. Case 1

The 26-year-old patient was married and worked part time in retail after discontinuation of her university studies due to the disorder. She had the first depressive episode in adolescence, initially diagnosed and treated as major depression. Two years later, she developed her first manic episode, and the diagnosis was corrected to bipolar I disorder. Two years before the pregnancy, a second manic episode occurred, for which she was again treated as an inpatient. After that only minor and moderate depressive episodes occurred, which were treated in the outpatient setting. Before she got pregnant, she was taking 150 mg sertraline, 90 mg amitriptyline oxide, and 1000 mg lithium carbonate per day. A secondary diagnosis was hypothyroidism that was continuously treated with L-thyroxine. Amitriptyline oxide was stopped 6 months before pregnancy because the patient was stable, and the pregnancy was planned. During the course of her pregnancy, she suffered from severe nausea but no hyperemesis and a light bleeding in the first 12 weeks. After that, the pregnancy was without complications, and the patient was mentally stable. In the 35th gestational week, a shortened cervix was diagnosed. Sertraline was continuously given as co-medication to lithium carbonate. Before and during the pregnancy, sertraline was given at 150 mg per day and then reduced in the 28th gestational week to 125 mg.

As shown in Table 1, lithium serum concentration decreased with stable dose by 40% in the first trimester, so that the dose was then increased by 30% until birth. After birth, the lithium dose was immediately decreased to the pre-pregnancy dose, which again resulted in therapeutic levels of lithium.

After having given birth to a healthy infant (female) after 37 weeks of pregnancy, the patient showed a mild euphoric mood, but this did not increase to a clinically significant hypomanic mood. The infant was born as a healthy, eutrophic, and mature newborn (APGAR 9/10/10, 2615 g (20. Percentile (P.), 46 cm (7. P.), head circumference 31.5 cm (4. P)). The infant did not show any postnatal adaptation syndrome and displayed normal thyroid values as well as no signs of a cardiac malformation. During the first 5 months, she showed normal growth and development. The infant was fully breastfed and at the age of 1 and 2 months, lithium levels were also measured in the serum of the infant. Here, 12.5% and 13.5% of mother’s serum lithium level could be detected (see Table 1). The infant’s thyroid values were also tested and were found to be in the normal range.

### 2.2. Case 2

The patient was 35 years old, was in a relationship, and was working as a teacher. She had been previously diagnosed with bipolar disorder and was already 10 weeks pregnant at initial presentation. She had stopped lithium medication abruptly after having discovered the pregnancy but then went back on because of severe mood swings. At age 25, she suffered from the first of a total of four manic phases and she had also experienced several depressive phases. Bipolar I disorder, however, was only diagnosed 5 years before the pregnancy, initially she had a diagnosis of borderline personality disorder. Secondary somatic diagnoses were reflux esophagitis, which was treated with pantoprazole, and hypothyroidism was treated with L-thyroxine.

Due to a quickly stabilized mood even with very low lithium serum levels below the therapeutic range after re-initiation of the lithium therapy, the lithium dose was not increased during pregnancy. In the 32nd gestational week, there was one unexpected increased lithium serum level of 0.55 mmol/L even when the lithium dose was stable; however, 4 weeks later, the lithium serum level was again very low with 0.16 mmol/L. Due to the already subtherapeutic lithium levels, we did not decrease the lithium dose further after birth, the serum levels increased slowly but were still subtherapeutic postpartum (see Table 2).

The pregnancy was without complications, labor was medically induced because there was no labor activity 10 days after the calculated birth date. The infant (male) was born as a healthy, eutrophic, and mature newborn (APGAR 9/10/10, 3450 g, 53 cm). The infant did not show any postnatal adaptation syndrome and displayed normal thyroid values as well as no signs of a cardiac malformation. He was breastfed in combination with formula during the first 4 months (about 50:50) and then the patient stopped breastfeeding. During the first 12 months, he showed normal growth and development. After 2 months, lithium levels were also measured in the serum of the baby, but those were below 0.1 mmol/L, therefore the measurement was not repeated. The infant’s thyroid and renal values were also tested and found to be in the normal range. Seven months pp, the patient discontinued lithium treatment because she felt well and the levels were below the therapeutic range anyway. The patient continued to be mentally stable and had a second pregnancy without medication 2 years later.

### 2.3. Case 3

The patient was 33 years old at initial presentation, married and working full time in civil service. The first depressive episode had occurred at age 25, subsequently and while taking antidepressant medication with venlafaxine, the patient developed her first and only manic episode, which led to the diagnosis of a bipolar I disorder. Before and at the time point when the patient got pregnant, she was mentally stable and treated with lithium carbonate LA (675 mg/d) and escitalopram (10 mg/d). Hypothyroidism was a known secondary diagnosis and medicated with L-thyroxine (100 µm) during pregnancy. In the first 12 weeks of pregnancy, she suffered from hyperemesis gravidarum. During the first weeks of pregnancy, she tried to stop the escitalopram medication, but it was again increased back to 10 mg because of recurring depressed mood. Lithium daily dose was decreased because of stable mood after the 13th gestational week to 450 mg, which still led to a serum concentration in the therapeutic range. In the following weeks, the lithium serum concentration decreased by 65% despite a constant dose. 

Around the 26th gestational week, the patient again reported more mood swings and more depressed mood, so the lithium dose was increased again by 225 mg every second day, which then led to a 40% increased lithium serum concentration in the 28th gestational week, however still below the therapeutic range. As the patient felt better, the dose remained stable. Unexpectedly, the patient then showed increasing lithium serum concentrations in the 31st and 39th gestational week, even after reducing the lithium dose, probably due to worsening renal function (see Table 3). Therefore, we further reduced the lithium dose shortly before birth to 225 mg, the patient then showed consistent low lithium serum levels but reported a stable mood, so the dose was not increased again. The infant (male) was then born healthy (height 50 cm, weight 2920 g) after 39 gestational weeks, and the patient started full breastfeeding. The infant did not show any postnatal adaptation syndrome and displayed normal thyroid values as well as no signs of a cardiac malformation. The infant was fully breastfed, and the first lithium measurement 11 days postpartum showed 42.3% of the mother’s serum concentration, which however was very low with 0.23 mmol/L (infant 0.11 mmol/L). The infant also showed normal development and no signs of the lithium exposure, so the mother continued breastfeeding. For 21 months, the infant showed normal growth and development and was first fully breastfed and still breastfed during the night until he was 21 months old. Lithium levels as well as renal and thyroid levels were measured every 4 weeks in the infant in the first 12 months (lithium levels in the infant were below 0.5 mml/L, exact data was not given by the pediatrician). After stopping breastfeeding, the patient experienced a short worsening in mood, which only lasted 3 days and improved without pharmacological intervention.

## 3. Discussion

When considering continuing a medication in pregnancy and the postpartum period, different risks and outcomes for the child and the mother have to be considered. Firstly, regarding the first trimester of the pregnancy, malformation risk is the most important outcome. Recently, it has been shown, that lithium exposure in the first trimester is associated with a malformation risk, but this is markedly lower than suggested formerly. The most recent meta-analysis reports an association of lithium exposure during early pregnancy with any congenital malformation (N = 23,300, odds ratio (OR) = 1.81, 95% confidence interval (CI) = 1.35–2.41; number needed to harm (NNH) = 33, 95% CI = 22–77). Regarding cardiac malformation, there was as well a significant but rather low risk associated with first trimester lithium exposure (N = 1,348,475, OR = 1.86, 95% CI = 1.16–2.96; NNH = 71, 95% CI = 48–167) [19]. None of the three infants in our report showed any congenital malformation; however, the malformation risk in general is very small with about 3% of all newborns involved and then only slightly increased to about 5% in lithium-exposed infants.

Secondly, it is of interest, whether medication during pregnancy is associated with pregnancy and birth complication. Here, data are scarcer; however, systematic reviews and meta-analysis report a higher risk that the infant needs to be admitted in neonatal care unit [6], but no significant associations with spontaneous abortion, preterm birth, low birth weight, or small for gestational age [5,19] were shown. Our three cases did not show any severe pregnancy or birth complication. With regards to the mothers, in our three cases we confirmed previous observations that lithium levels need to be monitored quite closely in pregnancy and around the time of the birth. As a result of an increased renal function in the first and second trimester, lithium levels in pregnancy decreased and slowly increased in third trimester and showed a more rapid increase after birth. Wesseloo et al., therefore, propose a monthly monitoring of lithium serum levels until 34th gestational weeks and a weekly monitoring thereafter until birth [6]. In our cases, we measured lithium levels every 4 weeks but did not measure weekly after the 34th week which was partly due to organizational issues. However, we also suggest implementing the weekly lithium measures in the clinical routine in the last weeks of the pregnancy because we could also see increasing lithium levels at stable doses shortly before and not only directly after birth.

Thirdly, the topics of potential fetotoxic effects and long-term effects on the neurodevelopment of the lithium-exposed children are of utmost importance. However, there are only few studies including small numbers of participants investigating this topic. Until now, no adverse long-term outcomes have been reported regarding developmental milestones and IQ (for a review see [16]). In our cases, we only have the reports of the mothers on the development up to 24 months on the three children. Here, no developmental abnormalities were reported.

Lastly, acute and long-term effects of medication exposure during breastfeeding should be considered. Breastfeeding is recommended by the World Health Organization (available online: https://www.who.int/health-topics/breastfeeding#tab=tab_1 (accessed on 5 March 2021)) for the first 6 months after birth as the optimal food for infants. It has repeatedly been shown that breastfeeding has health promoting effects not only in the children but also in the mothers [9,20,21]. Therefore, in general, women should be encouraged and supported to breastfeed their infants. However, if the women are in need of continuous medication, the risks and benefits need to be weighed. In the case of lithium, there is a still ongoing debate on whether breastfeeding should be recommended due to the excretion of lithium into the breastmilk. The mean of the reported infant levels in the previously published case reports and case series was about 25% [16]. In our three cases, the range was wide, between <2% and 43.2%, with the lowest concentration in the infant that was not exclusively breastfed, which strengthens the recommendation of regular measurement in the children. The three infants did not show any clinical symptoms or developmental abnormalities during the observed time period. However, we did not find it easily to collaborate closely with the treating pediatricians and obstetricians.

## 4. Conclusions

Consistent with previous recommendations, we suggest monthly lithium concentration measurements during the pregnancy until the 33rd gestational week and then weekly measurements until birth. Furthermore, we then recommend daily measurements as long as the patient is in the obstetrics ward (usually about three days) and at least one measurement in the infants directly after birth. After the infant has started to be fully breastfed, we recommend testing lithium level and renal and thyroid function in the infant after about one week and close clinical monitoring of the child with regard to weight gain, tremor, and vigilance. Depending on the lithium level in the infant’s blood after one week of full breastfeeding, the infant might be tested every 4 weeks as long as it is fully breastfed. If the infant’s lithium level is below 0.1 mmol/L at the first test, this might not be necessary to be repeated if the infant shows normal growth and development and the mother’s lithium level stays stable. However, for a more detailed recommendation, more data are needed. In the future, more efforts and potentially formalized networks with all involved professionals would be desirable to build a multidisciplinary support team for pregnant and postpartum bipolar women and their families.

## Figures and Tables

**Table 1 medicina-57-00634-t001:** Clinical data case 1.

Gestational Weeks	Lithiumcarbonat LA Daily Dose	Lithium Serum Levels [mmol/L]	Creatinine [mg/dL]	Renal Function, cGFR (MDRD) [mL/min/1.73 m^2^]
Before pregnancy	1000 mg (1-0-0-1½ 400 mg-tablets)	0.77	0.92	74
11th	1000 mg (1-0-0-1½ 400 mg-tablets)	0.47	N/A	N/A
13th	1000 mg (1-0-0-1½ 400 mg-tablets)	0.46	0.62	116
18th	1200 mg (1 ½ -0-0-1 ½ 400 mg-tablets)	0.52	N/A	N/A
20th	1400 mg (1 ½ -0-0-2 400 mg-tablets)	0.58	N/A	N/A
24th	1400 mg (1 ½ -0-0-2 400 mg-tablets )	0.56	0.70	101
28th	1400 mg 1 ½ -0-0-2 400 mg-tablets )	0.68	0.67	101
32nd	1400 mg (1 ½ -0-0-2 400 mg-tablets )	0.63	0.71	99
37th	1400 mg (1 ½ -0-0-2 400 mg-tablets )	0.76	0.72	97
Childbirth (38th gestational week)				
1 day pp	1000 mg (1-0-0-1½ 400 mg-tablets)	0.67	N/A	N/A
2 days pp	1000 mg (1-0-0-1½ 400 mg-tablets)	0.64	N/A	N/A
4 days pp	1000 mg 1-0-0-1½ 400 mg-tablets)	0.54	N/A	N/A
5 days pp	1000 mg (1-0-0-1½ 400 mg-tablets)	0.66	N/A	N/A
7 days pp	1000 mg (1-0-0-1½ 400 mg-tablets)	0.60	0.82	84
1 month pp	1000 mg 1-0-0-1½ 400 mg-tablets)	0.72 mother 0.09 infant Ratio infant maternal serum lithium: 12.5%		
2 months pp	1000 mg (1-0-0-1½ 400 mg-tablets)	0.89 mother 0.12 infant Ratio infant maternal serum lithium: 13.5%		

LA = long acting; pp = postpartum; N/A = not available; MDRD = Modification of Diet in Renal Disease; cGFR = calculated glomerular filtration rate.

**Table 2 medicina-57-00634-t002:** Clinical data case 2.

Gestational Weeks	Lithium Carbonate LA Daily Dose	Lithium Serum Levels [mmol/L]	Creatinine [mg/dL]	Renal Function, eGFR (MDRD) [mL/min/1.73 m^2^]
10th	450 mg (½-0-0-½ 450 mg-tablets)	0.29	N/A	N/A
18th	450 mg (½-0-0-½ 450 mg-tablets)	0.17	0.49	>120.0
21st	450 mg (½-0-0-½ 450 mg-tablets)	0.08	0.43	>120.0
32nd	450 mg (½-0-0-½ 450 mg-tablets)	0.55	0.55	>120.0
36th	450 mg (½-0-0-½ 450 mg-tablets)	0.16	0.43	>120.0
Childbirth (42nd gestational weeks)				
4 days pp	450 mg (½-0-0-½ 450 mg-tablets)	0.2	N/A	N/A
11 days pp	450 mg (½-0-0-½ 450 mg-tablets)	0.15	0.63	108.8
2 months pp	450 mg (0-0-0-1 450 mg-tablet)	0.42 mother <0.01 infant Infant was breastfed in combination with formula approximately 50:50	0.71	94.8
4 months pp	450 mg (0-0-0-1 450 mg-tablet)	0.21	0.61	113

LA = long acting; pp = postpartum; N/A = not available; MDRD = Modification of Diet in Renal Disease; cGFR = calculated glomerular filtration rate.

**Table 3 medicina-57-00634-t003:** Clinical data case 3.

Gestational Weeks	Lithium Carbonate LA Daily Dose	Lithium Serum Levels [mmol/L]	Creatinine [mg/dL]	Renal Function, eGFR (MDRD) [mL/min/1.73 m^2^]
Before pregnancy	675 mg (0-0-0-1 ½ 450 mg-tablets)	N/A	N/A	N/A
13th	675 mg (0-0-0-1 ½ 450 mg-tablets)	0.71	N/A	N/A
16th	450 mg (0-0-0-1 450 mg-tablet)	0.65	0.81	81.9
20th	450 mg (0-0-0-1 450 mg-tablet)	0.34	0.62	111.6
24th	450 mg (0-0-0-1 450 mg-tablet)	0.24	0.65	105.0
28th	450 mg/day alternating with 675 mg/day every other day (0-0-0-1/1½ 450 mg-tablets)	0.42	0.72	93.3
31st	450 mg/day alternating with 675 mg/day every other day (0-0-0-1/1½ 450 mg-tablets)	0.65	0.82	80.3
39th	450 mg (0-0-0-1 450 mg-tablet)	0.93	1.2	N/A
Childbirth (39th gestational week)				
5 days pp	225 mg (0-0-0-1/2 450 mg-tablet)	0.4	N/A	N/A
11 days pp	225 mg (0-0-0-1/2 450 mg-tablet)	0.26 Mother 0.11 Infant Ratio infant maternal serum lithium: 42.3%	N/A	N/A
3 months pp	225 mg (0-0-0-1/2 450 mg-tablet)	0.23 Mother <0.5 infant, exact value not available	0.83	N/A
6 months pp	225 mg (0-0-0-1/2 450 mg-tablet)	0.26	0.86	N/A
7 months pp	225 mg (0-0-0-1/2 450 mg-tablet)	0.25	0.8	N/A
8 months pp	225 mg (0-0-0-1/2 450 mg-tablet)	0.25	0.83	N/A
9 months pp	225 mg (0-0-0-1/2 450 mg-tablet)	0.14	0.81	N/A
10 month pp	225 mg (0-0-0-1/2 450 mg-tablet)	0.27	0.86	N/A
16 months pp	225 mg (0-0-0-1/2 450 mg-tablet)	0.25	0.80	N/A
19 months pp	225 mg (0-0-0-1/2 450 mg-tablet)	0.14	0.80	N/A

LA = long acting; pp = postpartum; N/A = not available; MDRD = Modification of Diet in Renal Disease; cGFR = calculated glomerular filtration rate.

## Data Availability

Due to data protection laws, we are not able to provide the clinical data of those patients.

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
