# Peer review of "Lithium Medication in Pregnancy and Breastfeeding—A Case Series"

_medicina, 2021, doi:10.3390/medicina57060634_

Round 1
Reviewer 1 Report
Provide brand name and manufacturer of the lithium products that each patient was taking and state if the daily dosage was given in all one dose daily or in divided doses. The reader needs to be able to assess the dosage form’s release characteristics and consequent fluctuation in maternal serum levels.
When were infant and maternal serum samples taken with respect to the mothers’ doses.
In case 2, please give the approximate percentages of breast milk and formula during the first 4 months.
Line 172: Describe the final daily doses rather than “225 mg every second day”. One possibility is “450 mg/day alternating with 675 mg/day every other day”
Line 178: change dosage to 225 mg/day.
Lines 243-244: It is inaccurate and misleading to state a value of <2% because this infant was not exclusively breastfed. If the infant received only 10% breastmilk, of course the serum level will be low.
Minor wording issues:
Lithium carbonate is 2 words – numerous instances
Line 52: change “as” to “than”
Line 102: change “sickness” to “nausea” or “nausea and vomiting” whichever is most appropriate.
Line 167: change “occurring” to “recurring” (if that is what is meant)
Line 187: change “full” to “fully”
Line 187: “after cost construction” makes no sense in English. Please reword or just leave it out.
Line 190: please explain what “exact data not available” means. Why? Were levels not recorded properly or were levels below the lower limit of quantification of the assay?
Line 202: change “as” to “than”
Line 212: change “data is” to “data are”
Lin 218: change “around the birth” to “around the time of birth”
Author Response
1.) Provide brand name and manufacturer of the lithium products that each patient was taking and state if the daily dosage was given in all one dose daily or in divided doses. The reader needs to be able to assess the dosage form’s release characteristics and consequent fluctuation in maternal serum levels.
We thank the reviewer for pointing out to those missing details. The requested information has been added to the revised manuscript.
All patients were taking Quilonum ret.® from Teofarma S.R.L. and GlaxoSmith Kline, those are the two manufacturer’s that are usually available. As this was a retrospectively analysed case series, we did not assess over that long period of time how many tablets from which of the two companies were taken by the patients. However, in our clinical experience, there is not a significant difference between the tables of those two manufacturer’s.
2.) When were infant and maternal serum samples taken with respect to the mothers’ doses.
3.) In case 2, please give the approximate percentages of breast milk and formula during the first 4 months.
The reviewer is right that this information was missing, we added it to the revised manuscript as follows: “Infant was breastfed in combination with formula approximately 50:50%”
4.) Line 172: Describe the final daily doses rather than “225 mg every second day”. One possibility is “450 mg/day alternating with 675 mg/day every other day”
We have changed the wording as the reviewer suggested in the revised manuscript.
5.) Line 178: change dosage to 225 mg/day.
We have added to the headings of the revised tables, that In any case, the daily dose of lithium medication is given to make it clearer as well as added the details on how the tablets were taken.
6.) Lines 243-244: It is inaccurate and misleading to state a value of <2% because this infant was not exclusively breastfed. If the infant received only 10% breastmilk, of course the serum level will be low.
The reviewer is right about that being inaccurate, thank you for pointing out, we deleted this in the revised table, also see answer to question 3.)
Minor wording issues:
- Lithium carbonate is 2 words – numerous instances
Done.
Line 52: change “as” to “than”
Done.
Line 102: change “sickness” to “nausea” or “nausea and vomiting” whichever is most appropriate.
Done
Line 167: change “occurring” to “recurring” (if that is what is meant)
Done.
Line 187: change “full” to “fully”
Done.
Line 187: “after cost construction” makes no sense in English. Please reword or just leave it out.
Done.
Line 190: please explain what “exact data not available” means. Why? Were levels not recorded properly or were levels below the lower limit of quantification of the assay?
The pediatrician did not report the exact data to the patient and to us, we have stated that now clearly in the revised version of the manuscript.
Line 202: change “as” to “than”
Done.
Line 212: change “data is” to “data are”
Done.
Line 218: change “around the birth” to “around the time of birth”
Done.
Reviewer 2 Report
Gehrmann and colleagues address a very important topic in their case report, namely pregnancy and breastfeeding while under Lithium medication. More and more women get challenged with the question to continue or to stop/switch medication when they want to become pregnant. Some even give up on their wish to have children when faced with this question. Therefore, case reports like the present are of great importance. Here, 3 extremely positive cases are reported. In addition, variety e.g. in terms of planned vs. unplanned pregnancy, full breastfeeding vs. supplementation with bottle is warranted. The manuscript is very well written and sufficient background information is given. I fully support publication of this article and have only minor comments.
- A closing statement in terms of recommendations could be really helpful. Authors mention the weekly lithium measures but do they have additional recommendations e.g. regarding birth or follow up checks on the children.
- Minor typos:
- Equal contribution of first authors is marked with # and not * (line 14)
- Line 27: repetition of ‘still’ can be avoided
- Line 192: first without should be deleted
Author Response
Gehrmann and colleagues address a very important topic in their case report, namely pregnancy and breastfeeding while under Lithium medication. More and more women get challenged with the question to continue or to stop/switch medication when they want to become pregnant. Some even give up on their wish to have children when faced with this question. Therefore, case reports like the present are of great importance. Here, 3 extremely positive cases are reported. In addition, variety e.g. in terms of planned vs. unplanned pregnancy, full breastfeeding vs. supplementation with bottle is warranted. The manuscript is very well written and sufficient background information is given. I fully support publication of this article and have only minor comments.
We thank the reviewer very much for the positive evaluation of our manuscript.
- A closing statement in terms of recommendations could be really helpful. Authors mention the weekly lithium measures but do they have additional recommendations e.g. regarding birth or follow up checks on the children.
We have added as much advice as we think can be given with regards to the sparse data available:
“Consistent with previous recommendations, we suggest monthly lithium concentration measurements during the pregnancy until the 33rd gestational week and then weekly measurements until birth. Furthermore, we then recommend daily measurements as long as the patient is in the obstetrics ward (usually about three days) and at least one measurement in the infants directly after birth. After the infant has started to be fully breastfed, we recommend to test lithium level, renal and thyroid function in the infant after about one week and closely clinically monitor the child with regards to weight gain, tremor and vigilance. Depending on the lithium level in the infant’s blood after one week of full breastfeeding, the infant might be tested every 4 weeks as long as it is fully breastfed. If the infant’s lithium level is below 0.1 mmol/l at the first test, this might not be necessary to be repeated if the infant shows normal growth and development and the mother’s lithium level stays stable. However, for more detailed recommendation, more data is needed.“
Minor typos:
- Equal contribution of first authors is marked with # and not * (line 14)
Done.
- Line 27: repetition of ‘still’ can be avoided
Done.
Line 192: first without should be deleted
Done.